# Group A Streptococcus Pili—Roles in Pathogenesis and Potential for Vaccine Development

**DOI:** 10.3390/microorganisms12030555

**Published:** 2024-03-11

**Authors:** Catherine Jia-Yun Tsai, Risa Takahashi, Jacelyn Mei-San Loh, Thomas Proft

**Affiliations:** 1School of Medical Sciences, University of Auckland, Auckland 1010, New Zealand; 2Maurice Wilkins Centre for Molecular Biodiscovery, Auckland 1010, New Zealand

**Keywords:** Group A Streptococcus, *Streptococcus pyogenes*, pili, vaccine

## Abstract

The Gram-positive human pathogen Group A Streptococcus (GAS, *Streptococcus pyogenes*) employs an arsenal of virulence factors that contribute to its pathogenesis. The pilus is an important factor that enables the pathogen to adhere to and colonize host tissues. Emerging research in pilus function shows that pili’s involvement in establishing infection extends beyond host adhesion. The diversity of GAS pilus types reflect the varying characteristics identified in different pili. With the development of new experimental systems and animal models, a wider range of biological functions have been explored. This brief review summarizes recent reports of new functions in different GAS pilus types and the methodologies that contributed to the findings. The established importance of the pilus in GAS pathogenesis makes this surface structure a promising vaccine target. This article also reviews recent advancements in pilus-based vaccine strategies and discusses certain aspects that should be considered in vaccine development according to the newly defined properties of pili.

## 1. Introduction

The pili of Group A Streptococcus (GAS, or *Streptococcus pyogenes*) were discovered literally with a touch of gold in 2005 via immunogold electron microscopy that used gold-conjugated antibodies that recognized the pilus components. These antibodies were originally raised against proteins in the Lancefield T-typing system and had been used for classifying GAS isolates for more than 50 years by that time [1]. GAS was among the first Gram-positive bacteria identified to express pili, although similar long, filamentous structures (pili or fimbriae) had been found on the cell surfaces of Gram-negative organisms many decades earlier. Since they were discovered, the number of research studies focusing on the structural, functional, and genetic features of GAS pili has rapidly and continuously increased.

The structural components and assembly enzymes of GAS pili are encoded within a genomic region characterized by a high level of recombination. This large (approximately 11–16 kb) region was later confirmed to be the location of genes for many previously identified surface proteins, including fibronectin-binding proteins (F1 [2] or SfbI [3] and F2 [4] or PFBP [5]), the collagen-binding protein Cpa [6], and the T antigen [7], therefore earning the name FCT (fibronectin binding, collagen binding, and T antigen) region. Increasing information on GAS genomics revealed the heterogeneity of this region, and so far, at least nine different FCT types distinguished by genetic composition and sequence arrangement have been designated [8]. The genetic divergence of FCT types reflects the antigenic variation in pili, part of which forms the basis of the T-serotyping scheme. All GAS strains carry and express one of the nine FCT regions, and this universal presence of the FCT region suggests the functional importance of the proteins encoded therein. In general, same-M-type strains express the same FCT subtype, but a single FCT type can be shared by multiple M-type strains [9,10,11].

The main component of the GAS pilus is the polymerized backbone pilin (BP), which has long been known as the T antigen, and one or two ancillary pilins (AP1 and AP2) [1]. AP1 was originally known as collagen-binding protein A (Cpa), which functions as the tip adhesin, while AP2 is the base subunit that serves as the cell wall linker [6,12,13]. The nomenclature used for each of the pilin proteins varies in the literature, and some common examples are provided in Figure 1. Earlier sequence analyses of the FCT region genes showed that BP-encoding genes can be grouped into 18 variants and 3 subvariants, and the *ap1* and *ap2* genes can be categorized into 14 and 5 variants, respectively [9,11], while more recent genomic epidemiology studies discovered even more variants [14]. Different from Gram-negative pili, GAS pili subunits are linked by covalent bonds to form a multimeric structure. AP1 and the BP each contain an LPXTG-like motif (e.g., VVPTG) in the C-terminus that is recognized by the pilus-specific sortase (SrtC). SrtC catalyzes the polymerization of pilins by forming amide bonds that join the carboxy group of the terminal threonine and the ε-amino group of a conserved lysine residue on an adjacent subunit. The process starts with the formation of an AP1-BP heterodimer which is then extended via the addition of BPs. The isopeptide bonds provide integrity for the long and thin fibrous structure, which can extend from the cell surface up to 10 μm. Additional structural integrity is achieved by intramolecular isopeptide bonds which are formed autocatalytically between lysine and aspartic acid/asparagine side chains [15]. The pilus fiber is extended until an AP2 subunit is added to the polymer [13,16]. AP2 contains the canonical LPXTG motif that cannot be recognized by SrtC, thus serving as the stop signal for pilus extension. However, the classical LPXTG motif is a substrate of the housekeeping sortase SrtA, allowing the assembled pilus to be linked to the amino side group of a specific amino acid within the cell wall peptidoglycan [17,18]. An exceptional case is the FCT-1 pilus of the M6 serotype strain, which lacks the base ancillary pilin, and the termination mechanism remains unknown. In this pilus, the pilus-specific sortase is SrtB, which belongs to the SrtC family and is responsible for the polymerization of the BP subunit (T6) as well as the linkage between T6 and the AP1 subunit FctX [19,20]. While all pili identified so far express the AP1 subunit, studies have shown that the deletion of AP1 in various FCT types does not abolish pilus assembly but reduces the polymerization efficiency of BPs [13,21,22].

The expression of pili is regulated by the RALP transcriptional regulators Nra or RofA in different strains [23,24,25,26]. In most cases, Nra and RofA act as positive regulators for pilus-related genes [27,28,29]. However, depending on the strain, Nra has been reported to have a dual effect on pilus expression [24,30]. Adding to the complexity, Nra and RofA are also regulated by other factors, including MsmR, [31], the two-component regulators [29,32], and even sRNA [33]. The dynamics of pilus expression have been linked to different stages of infection. It has been postulated that GAS downregulates pilus expression to allow for reduced bacterial adherence which, in turn, promotes the dissemination of the bacteria to other tissues [34]. Supporting this theory, Flores et. al. described a mutation associated with increased pilus expression that contributes to carrier phenotype and reduced virulence [35]. Although the molecular details of these regulatory mechanisms are yet to be defined, environmental cues that alter the pilus expression level have been reported, including temperature, pH, and oxygen content [23,25,36]. However, the biological significance of these types of environmental regulation remains to be established as these mechanisms are not widely adopted by all strains and seem to be highly serotype- or even strain-specific.

Since the discovery of pili in GAS, several biological functions and their associated importance in virulence have been investigated. This review aims to provide an update on the current understanding of pilus function and advances in the pilus research toolkit, as well as implications in vaccine development.

## 2. Roles in Pathogenesis

Many virulence factors have been described for GAS, a successful human-exclusive pathogen. The wide array of virulence determinants contribute to the ability of this bacterium to cause a large variety of diseases ranging from self-limiting pharyngeal or skin infections to life-threatening toxic shock syndrome and necrotizing fasciitis, as well as serious sequelae such as glomerulonephritis and rheumatic heart disease. Compared to other surface proteins, pili are unique because of their heterogenous composition. Furthermore, their covalently bound polymeric form also allows them to function away from the bacterial surface. These structural and physiological features provide them distinctive characteristics in the course of infection. Over the past decade or so, several studies have investigated the roles of GAS pili in pathogenesis. The diversity of functions reported for different pilus types (summarized in Table 1) mirrors their heterogeneity. Figure 2 and the section below provide an overview of the current understanding of pilus functions.

### 2.1. Adhesion

Components of pili belong to the extracellular matrix (ECM)-binding protein family, which has been shown to play an important role in host cell adhesion [44]. The adhesive property of pili was first confirmed by Abbot et. al. and Manetti et. al., using both primary tissues and immortal cell lines that represent clinically relevant GAS infection sites [38,40]. The original study by Abbot et. al. also implied that the adhesion determinant in the M1/T1 strain SF370 was the AP1 subunit (Spy0125), which was later confirmed by Smith et. al. using a *spy0125* deletion mutant as well as rSpy0125 antisera in an epithelial-cell-binding assay [16]. Following this, the adhesive properties of other pilus types were also described [22,39,43].

The adhesive characteristic of GAS pili has been attributed mainly to the tip subunit AP1, but a binding capacity of the backbone pilin has also been described in relatively atypical pilus types such as the FCT-6 pilus produced in the M2/T2 serotype. Using a GAS mutant devoid of AP1 as well as a gain-of-function *Lactococcus lactis* expressing the FCT-6 pilus without the tip subunit, research showed that the BP subunit is the main contributor to host cell adhesion and invasion and binds to host molecules including fibronectin and fibrinogen [22]. Coincidently, the BP of another less-studied pilus type produced in the M4/T4 serotype (FCT-5 pilus) has also been shown to bind the human inflammation marker haptoglobin [43]. The adherence characteristics of these BP proteins render them resistant to host bactericidal mechanisms, which will be discussed later in this review.

### 2.2. Biofilm Formation

The assembly of aggregates promotes bacterial survival and adherence to host tissues. In GAS clinical infections, the presence of biofilms suggests that the formation of microcolonies may play a role in GAS pathogenesis [45,46]. Different GAS strains exhibit differing propensities for biofilm formation, and there is even intra-strain variability within certain M types [46]. Research has shown that M1, M12, and M28 are often associated with a moderate level of biofilm formation, whereas M6 strains tend to produce greater biofilm mass [47]. Various surface virulence factors have been shown to contribute to GAS biofilm formation, including lipoteichoic acid (LTA), M protein, the HA capsule, collagen-like protein (Scl-1), and pili. Direct evidence of pili’s involvement in biofilm formation has been demonstrated in the FCT-1 (M6/T6), FCT-2 (M1/T1), and FCT-5 (M4/T4) strains [37,40,43], and again, the tip ancillary pilin was determined to be the key player [48]. Efforts have been made to verify correlations between certain FCT types and biofilm formation; however, a survey of 183 isolates grown under different conditions did not reach a conclusion [10].

For the FCT-1 pilus, the assembly of pili seems to be required for biofilm formation, as the deletion of any pilus subunits resulted in reduced biofilm formation in the M6/T6 strain TW3558 [37]. Interestingly, when an FCT-1 pilus operon originating from a M6/T6 strain was ectopically expressed in an M1/T1 serotype background, increased biofilm formation was seen [37], further confirming the unique role of this pilus in biofilm formation.

Apart from biofilm formation, pili-promoted GAS aggregation has also been reported in saliva. In a study by Edwards et al., the expression of diverse pilus types (FCT-1, FCT-2, FCT-3, and FCT4) mediated bacterial aggregation in saliva, which reduced bacterial adhesion to human epithelial cells [42]. Further investigation showed that the salivary component gp340 was responsible for this aggregation, which serves as a host defense mechanism.

### 2.3. Virulence

Studies of the relationship between GAS pili and virulence are often conducted in mouse infection models and via different forms of killing assays. The correlation between pilus expression and GAS virulence is highly variable among different FCT types and sometimes even contradicts within the same pilus type. In general, pili produced by the M2, M4, and M6 strains (the FCT-6, FCT-5, and FCT-1 types, respectively) have been shown to contribute to virulence [22,24,43,49]. In contrast, the FCT-2-type pilus expressed by the M1 strain was associated with reduced virulence in a mouse subcutaneous infection model as well as decreased bacterial survival in human blood [39]. The underlying mechanism was said to be the induction of neutrophil IL-8 production, followed by neutrophil endothelial transcytosis and eventually neutrophil extracellular traps (NETs), which promote bacterial entrapment and killing [39].

Research on the FCT-3 pilus, which is the most common form of the FCT region, produced the most conflicting results regarding pili’s correlation with virulence. FCT-3 pilus expression in M3 GAS was shown to promote bacterial killing in human blood [27]. Interestingly, the abundance of pili in the M3 strain was found to be significantly lower than other serotypes, and this lack of pilus expression was hypothesized to be a contributing factor in the more frequent appearance of M3 isolates in highly invasive infections [27]. However, the same FCT-3 pilus expressed in an M53 serotype background has been associated with increased virulence in a human-skin-engrafted SCID mouse model [21]. Similarly, in the M49 serotype background, reduced transcription of the FCT-3 locus resulting from the deletion of the MsmR regulatory factor presented a less virulent phenotype [31].

Of note, the definition of “virulence” itself can sometimes contribute to variability. While the virulence of a microorganism often refers to its ability to infect and cause damage to a host, the enhanced adherence (and thus an increased ability to infect) promoted by pilus expression is sometimes associated with a more harmless carrier phenotype.

### 2.4. Immune Activation or Evasion

The surface-abundant GAS pili are no doubt exposed to the host immune system; however, relatively little information is known about the interactions between GAS pili and immune components. Early research mainly focused on the adhesive property of pili, but evidence of immunological characteristics have been demonstrated in the well-studied FCT-2 pilus expressed by M1 strains. The pili of the M1T1 strain 5448 have been shown to stimulate the production of pro-inflammatory IL-8 and DNA-based extracellular traps (NETs) in neutrophils [39]. Similarly, fully assembled pili as well as recombinant pilus proteins from the M1 SF370 strain stimulated the release of TNF and IL-8 in human monocytic THP-1 cells, and this pilus type was further confirmed to be a ligand of toll-like receptor 2 (TLR2) [41]. A follow-up study investigated the pro-inflammatory activity of different types of GAS pili and concluded that pilus-mediated inflammation is likely an immune alarm system that promotes bacterial clearance rather than causing disruptive damages associated with disease manifestation [50]. This conclusion also suggests that the immune-potentiating property of pili may be exploited in adjuvant development.

Apart from host cell adhesion, immune evasion has also been suggested as a mechanism that contributes to pilus-mediated virulence. Interestingly, the less conventional pilus types have provided more molecular details on this aspect. The FCT-6 pilus exclusively produced in the M2 serotype is partially orthologous to the PI-1 pilus of Group B Streptococcus (*Streptococcus agalactiae*). The two ancillary pilins share high amino acid sequence identity with GBS minor pilins, but the BP protein is only 42% identical to the GBS major pilin and also lacks significant homology to any other known pilus backbone protein. This BP (Spy0109) also exhibits unique features in immune evasion, demonstrated by its interaction with monocytes, reduced macrophage uptake and killing, as well as resistance to whole blood killing [22]. Although the mechanisms underlying this unusual immune evasion function are yet to be defined, the protein’s interaction with human extracellular matrix proteins such as fibrinogen may play a role. The fibrinogen binding of other GAS surface components such as the M protein has been associated with enhanced invasiveness [51,52]. The FCT-6 BP may adapt a similar mechanism, or fibrinogen binding may lead to local fibrin aggregation that enables immune evasion via molecular camouflage. On the other hand, the haptoglobin binding of the FCT-5 BP was suggested as a previously unappreciated virulence-enhancing mechanism of pili [43], as haptoglobin is upregulated during acute infection and the surface sequestration of this serum-abundant protein by GAS may mask the bacterium from immune surveillance [53].

## 3. Experimental Systems for Functional Characterization

The early functional characterization of GAS pili was mostly conducted using in vitro cell culture systems. Pharyngeal cells such as Detroit562 or HEp-2 and keratinocytes such as HaCaT are among the most commonly used cell lines for studying adherence, invasion, and microcolony formation as they represent the primary infection sites of GAS [16,39,40]. However, discrepancies in results from freshly isolated primary cells versus immortal cell lines have been noted, emphasizing the limitation of using lab-grown monolayer cells that do not mimic the squamous epithelia of the skin or throat [38].

Another in vitro system used to characterize the biological function of GAS pili is the Lancefield or whole blood killing assay. The classical assay, named after the prominent streptococcal researcher Rebecca Lancefield, was initially developed as a protocol for identifying novel GAS serotypes [54,55]. The modified method has been widely used in the study of many GAS virulence factors, including the pili of the M1, M2, and M4 strains [22,39,43], and usually correlates well with in vivo virulence in animal models. Apart from whole blood, human saliva [56], macrophage cell lines [22,41], and antimicrobial peptides [41] have also been used to assess the fitness of pilus-deletion mutants.

Direct evidence of the pilus proteins’ involvement in colonization and virulence can be demonstrated in animal models. An early example is the superficial skin infection model using humanized mice, which showed attenuated virulence in a Cpa-deletion mutant [21]. On the contrary, Nakata et al. noted increased virulence associated with disrupted pilus assembly using a dermonecrotic mouse infection model [13]. The contradicting results are likely due to the different experimental setups used for animal models, which adds another layer of complexity to the existing challenges in establishing an animal model for this human-exclusive pathogen. Previous studies have shown that most GAS strains colonize poorly in mice, but an M75 strain, an M89T11 strain, and an M49T18.2 strain have been shown to cause infection in mice more efficiently [57,58,59]. Advanced genetic technologies also enabled the creation of humanized transgenic mouse strains that are useful for studying GAS virulence, host–pathogen interactions, and vaccine efficacy [60].

An alternative infection model for GAS and other pathogenic bacteria coming under the spotlight in recent years is the *Galleria mellonella* larvae model [61,62]. The inoculation of wildtype or non-piliated GAS mutants resulted in measurable differences in a comprehensive health index scoring system. Bacterial burden, determined either by plating enumeration or the use of bioluminescent-labeled bacteria, can also be used to measure the virulence of a GAS strain. To quantify more instant and somewhat more subtle infection outcomes, the numbers of hemocytes isolated from the larvae hemolymph can be used [50].

Modified protocols of the Lancefield whole blood killing assay have been routinely used as classical methods for screening the bactericidal activity of vaccine antisera [63]. To overcome issues associated with the variability and accessibility of human whole blood, modern opsonophagocytic killing assays have been developed for the evaluation of GAS vaccine antisera [64,65,66]. These methods use dimethylformamide (DMF)-differentiated human promyelocytic leukemia cells (HL-60) as a proxy for neutrophils, hereby reducing the primary source of variation in the classical assay.

## 4. Potentials in Vaccine Development

Apart from functional characterization, vaccine development is no doubt a popular theme in pilus research. The fact that the major component of GAS pili is a key antigen lays the foundation for a vaccine based on these pili. More recently, the pilus structure has been used in a number of innovative vaccine platforms for antigen delivery. Emerging structural and immunological analyses of T antigens provide new insights that can inform the design of pilus-based vaccine strategies. For example, epitope identification using a Fab-phage display technology provided molecular details that form the basis of T antigen cross-reactivity [67]. A recent study clarifies that the inflammatory, stimulating effect of GAS pili serves as an immune-activating signal rather than the cause of clinical presentations, therefore supporting the use of the pilus in vaccine development [50]. This section describes a few examples of vaccine innovation based on GAS pili, and Figure 3 below provides a schematic illustration of different vaccine strategies.

### 4.1. Pilus-Based GAS Vaccines

GAS continue to cause significant health burdens worldwide. Superficial infections such as pharyngitis and impetigo affect hundreds of millions people each year, while more severe conditions cause more than 500,000 deaths per annum [68,69,70]. The greatest burden is attributed to rheumatic heart disease (RHD), which often develops after acute rheumatic fever (ARF), with a global prevalence of approximately 15 million [69,71]. Although ARF and RHD rates have generally been decreasing in most parts of the developed world, they remain high in low-income countries [72]. In addition, disproportionally high rates of ARF and RHD continue to be reported in specific populations in higher-income countries, for example, the indigenous peoples of New Zealand and Australia [73,74]. Despite this alarming evidence of a lack of effective disease control, so far, there is no licensed vaccine against GAS. Currently, there are only four GAS vaccines in clinical development which are all based on the M protein [75,76,77,78]. The M protein is a well-studied virulence factor of GAS and the basis of an alternative Lancefield serotyping scheme. Early vaccine development utilizing crude M protein preparations was associated with adverse ARF events [79]. This unfortunate consequence was speculated to be the result of similar epitopes presented in the M protein and human heart tissue [80]. Current efforts mostly focus on subunit strategies, but debates remain on whether to use the more immunogenic but highly variable N-terminus or the conserved C-repeat region that may have issues regarding immune accessibility.

Pili present a promising alternative to the M protein-based vaccine strategy. Studies of Gram-negative pili established the importance of pili in virulence and have paved the way for pilus-based vaccine development. The prospects of a pilus-based vaccine for GAS disease were immediately recognized upon the discovery of this surface structure. An early study by Mora et. al. showed that immunization with a combination of the three recombinant pilus subunits of the M1 SF370 strain conferred >70% protection against a nasal challenge with a mouse-adapted M1 strain [1]. In general, the backbone and ancillary pilins are both immunogenic, but considering the accessibility and abundance of the antigen in the assembled form of pili, the BP (T antigen) has been the focus of pilus-based subunit vaccine development. However, fully assembled pili expressed on the probiotic *Lactococcus lactis* have been demonstrated to be a novel means of producing a self-adjuvating pilus-based vaccine [59]. *Lactococcus* and *Streptococcus* are related Gram-positive organisms that use sortase A for surface protein display, which allows for the heterogeneous expression of the GAS pili in a *Lactococcal* surrogate strain transformed with the *Streptococcal* pilus operon.

Lack of specificity is a known drawback of the T serotyping scheme. However, this drawback becomes an advantage for a T antigen-based vaccine since the existence of cross-reactivities can increase vaccine coverage. The fact that there are far fewer T types compared to M types (approximately 20 *tee*-types vs. more than 200 *emm*-types) also means that a broader strain coverage can be achieved more easily by a vaccine based on the T antigen. In addition, structural studies showed that despite moderate sequence similarity, several T antigens (T3.2, T13, and T18.1) shared remarkable structural similarity to the well-studied T1 [81]. This structural conservation and shared epitopes between T antigens provide the basis of the cross-reactive antibody response induced by T antigens. Indeed, Khemlani el. al. demonstrated that a combination of seven *L. lactis* strains, each expressing a different GAS pilus type, induced cross-reactive antibodies that recognized non-vaccine strains [82]. A recent study by Loh et. al. also showed that the administration of three chimeric recombinant proteins consisting of multiple domains from different T antigens produced cross-reactive antibodies in rabbits that are expected to provide >95% vaccine coverage [83]. Importantly, the antibody titers of specific T antigens elicited from immunization with an individual T antigen or with the multivalent recombinant protein were similar, which translated to comparable levels of protection in a mouse invasive challenge model. Although the discovery of new T antigen variants in a more recent genomic epidemiology analysis has reduced the vaccine coverage to ~80%, the estimated coverage of TeeVax is still higher than the 30-valent M protein vaccine in Pacific children [14].

### 4.2. GAS Pili as a Vaccine Carrier

The pilus structure heterologously expressed on the surface of *L. lactis* has also been used as a carrier for vaccine delivery. Pili extend a long distance from the cell wall, providing B cells or antigen-presenting cells (APCs) access to the antigen. The first design of such an innovation consisted of an HIV antigen incorporated into the tips of T3 pili expressed by *L. lactis* which successfully induced antigen-specific systemic and mucosal antibody responses in orally immunized mice [84]. The study also demonstrated that the pilus carrier activates dendritic cells (DCs) in the small intestine, which might have contributed to the strong mucosal immunity induced by the vaccine [85].

As opposed to using the pilus tip to carry a recombinant antigen, another strategy termed “PilVax” utilizes the pilus backbone as a carrier for a peptide antigen [86]. The intra- and intermolecular peptide bonds of pilus subunits as well as the covalent linkages between assembled pilins confer strong structural stability and protect incorporated peptides from proteolysis. The additional advantage of this strategy is that through the polymerization of the backbone pilin, amplification of the incorporated antigen can be easily accomplished. A pilot study demonstrated that intranasal immunization with the prototype PilVax construct containing a model peptide from ovalbumin elicited substantial serum IgG and mucosal IgA antibodies [86]. Subsequent studies showed that a tuberculosis vaccine carrying the *Mycobacterium tuberculosis* Ag85b T cell epitope induced antigen-specific CD4^+^ T cell responses in which the levels were similar to those induced by the benchmark vaccine BCG [87]. In addition, a PilVax vaccine targeting *Staphylococcus aureus* successfully protected intranasally immunized mice against the mouse-adapted *S. aureus* strain JSNZ [88]. However, the study also reported that vaccination with a control PilVax strain without the antigen also significantly reduced bacterial load in the infected mice [88]. Since the pilus has been shown to induce strong pro-inflammatory cytokines [41], this suggests that the immune-stimulating property of pili may result in innate immune priming that heightens non-specific defense mechanisms.

## 5. Discussion

The identification of the FCT region opened up an era of the genetic characterization of this locus as well as the functional characterization of the proteins encoded in this genomic region. The highly diverse FCT regions correspond to the heterogeneity of the factors expressed which, in turn, contributes to highly variable phenotypes such as host cell interactions and biofilm formation [10]. Apart from the primary function of host cell adhesion, new biological roles have been identified, including immune evasion and stimulation. With the advancement in new molecular biology tools and animal models, one can expect new information to continue to be discovered for the pilus. Pathogenomic studies have reported several cases in which the altered expression of pili contributed to a change in the pathogen’s virulence, confirming the importance of pili in pathogenesis proposed by in vitro and animal studies in the past. Recent global genomic analyses revealed that circulating strains in low-income countries are more diverse than in high-income countries, which is also reflected in the diversity of the FCT types expressed in prevalent strains in certain regions [89]. Pili remain an important epidemiological marker, but this implication has become more complicated than ever due to the increasing information gathered thus far. There have been several theories on the contribution of pili to tissue tropism, disease manifestation, and clinical consequences, but these still require confirmation with experimental evidence.

As the regulatory mechanisms affecting pilus expression vary between strains, levels and patterns of pilus expression among different strains are far from uniform [27,90,91]. The accessibility of pili can also be masked by other surface components, such as the hyaluronic acid (HA) capsule. The HA capsule has long been shown to interfere with GAS adhesion to various cell types [92,93,94] and was thought to be the cause for the lack of pilus-specific antibody-mediated neutralization seen in a hyper-encapsulated M18T18.1 strain [59]. The regulation of pilus expression and the biosynthesis of other surface components are intricate and add complexity to research on the biological and clinical implications of pili. These dynamics mean that even for the same pilus type, there might be variation in its interactions with host molecules which subsequently influence the virulence and fitness of GAS.

The discovery of new structural and biological functions of pili has potential for many innovations. For example, the unique intramolecular isopeptide bond of the backbone pilus protein may be exploited in a wide range of applications, such as the ones shown by the SpyTag toolbox [95]. Methodology for engineering an isopeptide bond into a predetermined immunoglobulin-like protein fold has also been published and can be adopted as a protein-stabilization strategy in fields such as antibody engineering [96]. The combination of the immune-stimulating pilus and the *L. lactis* vehicle opens a new avenue for vaccine development and can be further developed into a versatile platform similar to the Mimopath technology [97]. The notion that pilus proteins are ligands of the immune receptor TLR2 suggests that pilus proteins can be developed as new adjuvants. The advantage of a protein-based adjuvant lies in the ease of producing an antigen–adjuvant complex by molecular cloning. A recombinant protein containing a pilus protein conjugated to an antigen ensures the efficient co-delivery of the adjuvant and antigen. Further investigation into the immunological features of pili may also lead to new applications in immune modulation or prophylactics.

GAS infections continue to cause high morbidity and mortality rates, especially in underprivileged settings. Progress toward an effective vaccine has been hampered by past failure and potential safety concerns. The suspected link between M proteins and rheumatic heart disease encouraged the exploration of alternative vaccine targets such as pili [98]. However, phase variation and antigenic variation present major hurdles for pilus-based vaccine development. Due to the highly dynamic nature of pilus expression, pilus status in both experimental and epidemiological contexts needs to be considered in the work of vaccine development. As several environmental factors have been associated with pilus gene regulation, culture conditions may alter the expression levels of pili, thus affecting the results of vaccine evaluation. The notion of thermosensitive and acid-induced pilus expression implies that at least for some strains, mucosal immunity induced by a pilus-based vaccine may exert better neutralizing activity than systemic immunity. This is because the generally lower temperature and acidity on the mucosal surface is more in favor of pilus expression for at least some strains [13,23,36]. Therefore a mucosally delivered pilus vaccine is expected to induce secreted antibodies that can effectively target piliated bacteria.

Pilus-based vaccines should offer sufficient coverage in order to avoid vaccine failure due to lack of cross-reactivity or due to potential vaccine-induced serotype replacement. However, creating a universal vaccine that covers all GAS strains may be an overambitious goal. Genomics studies to characterize the regional GAS population are thus important to inform effective vaccine design. Lacey et al. recently performed such an investigation for New Zealand and reported that although circulating strains in New Zealand show a similar pattern as those in low- and middle-income countries, strains commonly seen in high-income countries that are associated with outbreaks and antimicrobial resistance were also observed [14]. On the other hand, Bah et al. analyzed and compared the molecular characteristics of circulating strains in a low-income country (Gambia) and a high-income country (UK) and concluded that while several molecular markers are shared, the *emm* type diversity and the prevalence of certain virulence factor genes differ between the two settings [89]. As *tee*-typing is historically less explored than *emm*-typing in GAS epidemiology, more regional and global efforts in circulating tee-type surveillance are required to sufficiently inform the development of T antigen-based vaccines.

## Figures and Tables

**Figure 1 microorganisms-12-00555-f001:**
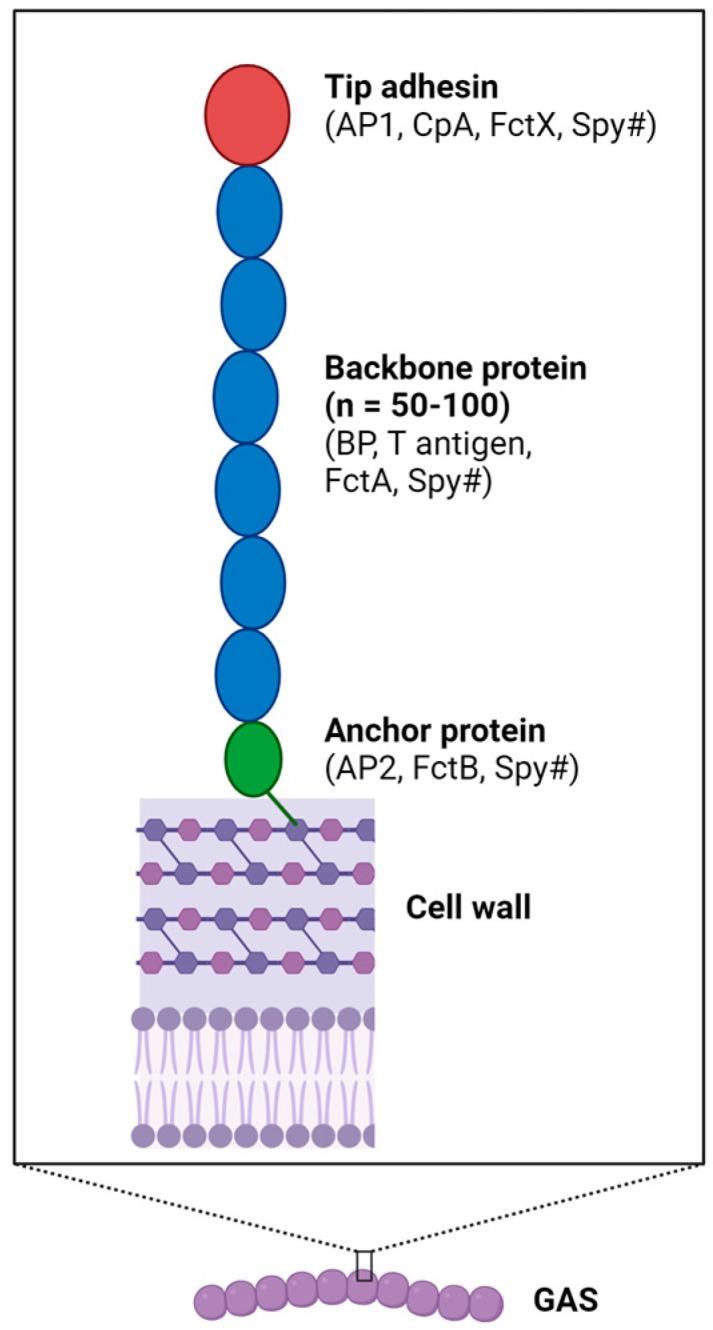
**Components and structure of pilus.** The three-component pilus comprises polymerized backbone proteins, a tip adhesin and an anchor protein. Different naming systems exist. For example, in the FCT-2 pilus produced by the M1T1 strain SF370, the BP is also called T1, FctA, or Spy0128; AP1 is also known as CpA or Spy0125; and AP2’s alternative names include FctB and Spy0130. Image created using BioRender.com.

**Figure 2 microorganisms-12-00555-f002:**
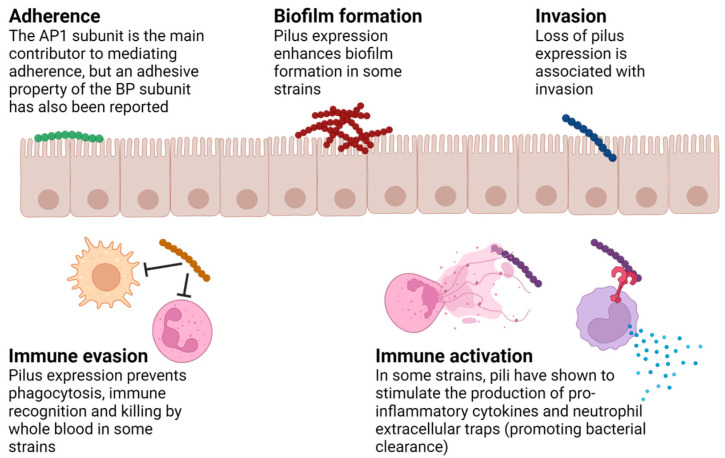
**Various functions reported for GAS pili.** Over the past decade, several biological functions of GAS pili have been described, including host cell adherence and invasion, immune evasion and activation, and biofilm formation. Image created using Biorender.com.

**Figure 3 microorganisms-12-00555-f003:**
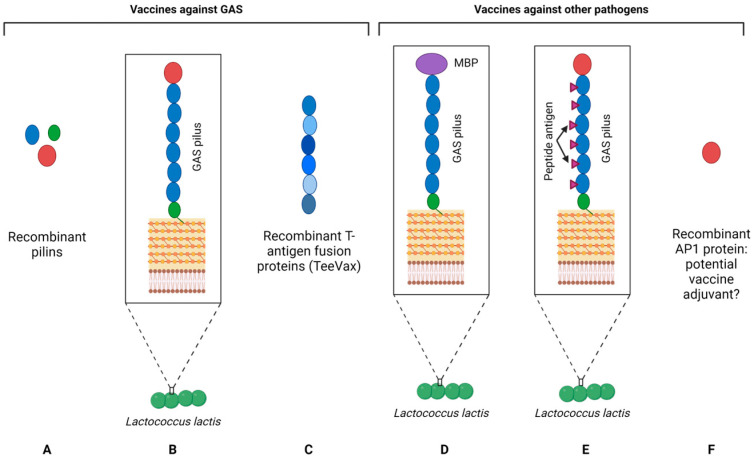
**Vaccine strategies based on GAS pili.** For a vaccine developed against GAS infection, either individual pilin recombinant proteins (**A**) or assembled pili expressed on *L. lactis* (**B**) have been shown to induce protective antibody responses. TeeVax (**C**) is an innovative T antigen-based GAS vaccine which comprises various domains of different T antigens. A total of 3 TeeVax recombinant proteins have been generated to provide a high level of vaccine coverage based on cross-reactivity among different T antigens. The pilus structure has also been exploited in vaccine platform development. An antigen can be inserted in place of the AP1 subunit (**D**) or within the BP protein structure (**E**). Furthermore, due to its potent immune-stimulating effect, recombinant AP1 has promise in the development of a novel vaccine adjuvant (**F**). Created using Biorender.com.

**Table 1 microorganisms-12-00555-t001:** Involvement of GAS pili in different stages of infection.

FCT Type	Stages of Infection	Determination of Characteristics	References
FCT-1	Adhesion	AP1 deletion reduces binding to epithelial cells	
	Biofilm formation	Loss of biofilm formation in BP and AP1 deletion mutants	[37]
	Virulence	AP1 deletion decreases GAS virulence in humanized mouse impetigo model	
FCT-2	Adhesion	Reduced binding to epithelial cells with AP1 deletion and with addition of anti-AP1 sera	[16,38,39,40,41]
	Biofilm formation	Loss of biofilm formation in BP and AP1 deletion mutants	
	Immune stimulation	AP1 and BP activate TLR2 and induce pro-inflammatory cytokine release	
FCT-3	Adhesion	Pilus deletion reduces binding to host proteins	
	Virulence	AP1 or BP deletion decreases GAS virulence in human-skin-engrafted mouse model	[21,27,31]
FCT-4	Adhesion	Pilus expressed on *L. lactis* binds to host proteins	[42]
	Biofilm formation	pH-dependent increase in pilus expression levels linked to biofilm formation	
FCT-5	Adhesion	Reduced binding to epithelial cells following BP deletion	[43]
	Biofilm formation	Loss of biofilm formation in BP deletion mutant	
	Virulence	BP deletion decreases GAS virulence in systemic mouse infection model	
	Immune evasion	BP mediates haptoglobin binding, which promotes reduced susceptibility to antimicrobial peptides	
FCT-6	Adhesion	BP mediates adherence to and invasion of epithelial cells and directly binds to fibrinogen	[22]
	Immune evasion	BP binds to monocyte, interferes with blood clot formation, and reduces macrophage uptake and killing	
	Virulence	Expression of pili provides resistance in whole-blood killing assay and a *Galleria mellonella* infection model

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
