# Peer review of "Group A Streptococcus Pili—Roles in Pathogenesis and Potential for Vaccine Development"

_microorganisms, 2024, doi:10.3390/microorganisms12030555_

Round 1
Reviewer 1 Report
Comments and Suggestions for Authors
Dear authors
Thanks for your nice work and presentation.
Few points should be considered during revision;
- The title could be “Group A Streptococcus pili – roles in pathogenesis and potentials in vaccine development in humans”.
- The references should be updated till 2023. Most of references are old.
- The aim of the work have to be added at the end of the introduction section.
- Table 1, the 2nd column “stage of infection“ is repeated in each row.
- The conclusion and recommendations should be added after discussion.
Best wishes
Author Response
We thank the reviewer for their constructive feedback. I'd like to respond to the comments as below:
- The title could be “Group A Streptococcus pili – roles in pathogenesis and potentials in vaccine development in humans” We thank the reviewer for the suggestion. However, we believe that adding "in humans" in the title may be limiting and unnecessary. For example, the PilVax platform is also suitable for the development of vaccines against infectious diseases in animals.
- The references should be updated till 2023. Most of references are old. We are not sure which references the reviewer is referring to. We believe that our references are up-to-date which is also acknowledged by the other two reviewers. There are 8 references from 2023 and one published in 2024.
- The aim of the work have to be added at the end of the introduction section. Thank you for the suggestion. We have added this at the end of the introduction section.
- Table 1, the 2nd column “stage of infection“ is repeated in each row. We believe this might be an error in the PDF format. We have checked that there is no erroneous repetition in the table.
- The conclusion and recommendations should be added after discussion. Thank you for the suggestion. We have incorporated our recommendations and conclusions in the discussion section, weaving in the text where appropriate.
Reviewer 2 Report
Comments and Suggestions for Authors
In this review the authors have summarized the structure and function of different GAS pilus types as well as their role in GAS pathogenesis and vaccine development in a clear and comprehensive way, citing recent and relevant articles.
Author Response
Thank you for your encouraging feedback!
Reviewer 3 Report
Comments and Suggestions for Authors
The authors of this review describe the pili of Group A Streptococcus (GAS). In their introduction, they describe the structure and assembly and the regulation of GAS pili. They then move to the roles of pili in GAS pathogenesis and in the interactions of the bacteria with their host.
Next, they report on the experimental systems used for functional characterization and discuss the limitations of animal models to study a strictly human pathogen. They then talk about vaccine development, which is a major theme in pilus research. In addition to strategies for Immunization against GAS (recombinant Lactococcus lactis, T-antigen based vaccines), their presentation includes the use of GAS pili as platforms for vaccines against other pathogens and the use of the AP1 protein as an adjuvant.
They have a good, up-to-date list of references. They do not shy away from exposing contradictions in the literature on GAS pili, which is a purpose of reviews. The figures are fine.
It is unusual to have a discussion section in reviews. I would rather suggest to discuss the various issues in the corresponding paragraphs and have a shorter conclusion.
Other than that, I have only minor comments.
1° Could the authors explain the difference between ‘higher levels of biofilm formation’ and ‘greater biofilm mass’ ?
2° Lines 92-94, add ‘that’ before contributes
3° Lines 185-187, replace resulted by resulting
4° Line 181: replace less by lower
5° Line 222 : enables
Comments on the Quality of English Language
I pointed a few grammatically incorrect sentences that need to be revised. Other than that the English is fine.
Author Response
We thank the reviewer for their constructive feedback. We agree that some review papers incorporate the various discussion points in corresponding sections without having a final discussion at the end. We did consider this format while writing the manuscript but found it harder to achieve a better flow than the current form. This is because we aim to discuss the overall biological and pathological importance of pili across the various functions, and also to point out some considerations in vaccine development based on the different features of pili. Therefore we still believe that a more integrative discussion will serve the purpose better.
Below is our response to the other comments:
1° Could the authors explain the difference between ‘higher levels of biofilm formation’ and ‘greater biofilm mass’ ?
Thank you for raising this question. It was indeed a typo. We have changed the "higher level" to "moderate level" in this sentence (Lines 148-149).
2° Lines 92-94, add ‘that’ before contributes
Added (current line number: 93).
3° Lines 185-187, replace resulted by resulting
Corrected (current line number: 190).
4° Line 181: replace less by lower
Updated (current line number: 185).
5° Line 222 : enables
Corrected (current line number: 226).